# Effect of Polycan, a β-Glucan from *Aureobasidium pullulans* SM-2001, on Inflammatory Response and Intestinal Barrier Function in DSS-Induced Ulcerative Colitis

**DOI:** 10.3390/ijms241914773

**Published:** 2023-09-30

**Authors:** Hyun Ju Do, Young-Suk Kim, Tae Woo Oh

**Affiliations:** 1New Drug Development Center, Daegu-Gyeongbuk Medical Innovation Foundation, Daegu 41061, Republic of Korea; dododo@dgmif.re.kr; 2Glucan Co., Ltd., 25-15, Worasan-ro 950 beon-gil, Munsan-eup, Jinju-si 52840, Gyeongsangnam-do, Republic of Korea; kys0725@glucan.co.kr; 3Korean Medicine (KM)-Application Center, Korea Institute of Oriental Medicine (KIOM), Daegu 41062, Republic of Korea; 4Department of Korean Convergence Medical Science, University of Science & Technology (UST), 1672 Yuseongdae-ro, Daejeon 34054, Yuseong-gu, Republic of Korea

**Keywords:** *Aureobasidium pullulans*, β-glucan, ulcerative colitis, intestinal battier, inflammation

## Abstract

Ulcerative colitis (UC), a subtype of inflammatory bowel disease, is a chronic gastrointestinal inflammatory disease with unclear etiology and pathophysiology. Herein, we determined the effects of extracellular polysaccharides purified from *Aureobasidium pullulans* SM-2001 (Polycan) on tight junction protein expression, inflammation, and apoptosis in a dextran sodium sulfate (DSS)-induced acute colitis model. Fifty mice were divided into normal, DSS, DSS + Polycan 250 mg/kg (Polycan 250), DSS + Polycan 500 mg/kg (Polycan 500), and DSS + 5-aminosalicylic acid 100 mg/kg (5-ASA) groups. Their body weights, colon lengths, histological changes in colon tissue, and tight junction function were observed. Results showed that Polycan 250, Polycan 500, and 5-ASA significantly inhibited body weight loss compared with DSS. Similar to 5-ASA, Polycan 500 exhibited preventive effects on colon length shortening and histological changes in colon tissues. Polycan inhibited the DSS-induced decrease in fluorescein isothiocyanate-dextran permeability and myeloperoxidase activity. Moreover, Polycan significantly recovered serum cytokine (e.g., tumor necrosis factor-α, interleukin (IL)-6, and IL-1β) or mRNA expression in colon tissue compared with DSS. Polycan also inhibited apoptosis by reducing caspase-3 activity and the Bcl-2 associated X/B-cell lymphoma 2 (Bcl-2) ratio. Additionally, DSS treatment significantly reduced microbial abundance and diversity, but the administration of Polycan reversed this effect. Collectively, Polycan protected intestinal barrier function and inhibited inflammation and apoptosis in DSS-induced colitis.

## 1. Introduction

Inflammatory bowel disease (IBD) refers to chronic inflammation of unknown etiology that occurs in the bowel and usually refers to the idiopathic inflammatory growth disorders ulcerative colitis (UC) and Crohn’s disease (CD), including the relatively common Bechet colitis [1,2]. In a broad sense, infectious enteritis such as bacterial, viral, amoebic, and tuberculous enteritis; ischemic enteritis; and radiation enteritis are all inflammatory diseases occurring in the intestine. The symptoms of colon inflammation vary depending on the extent of the lesion and the degree of inflammation, but the most conspicuous are changes in defecation habits such as diarrhea and bloody stools [3]. Patients with UC have a two- to eightfold higher incidence of colorectal cancer than those without. UC is common in Western countries, such as the United States, and Europe. Although the cause of UC remains unknown, the interaction of internal factors (e.g., genes) and external factors (e.g., diet, alcohol consumption, tobacco, and obesity) is believed to induce colitis [4]. Regarding the mechanism by which UC develops into cancer, proinflammatory cytokines activate the defense response of immune cells (e.g., lymphocytes, macrophages, and neutrophils) in colonic mucosa, leading to inflammatory responses. Many studies have been published showing differences in gut microbial diversity between healthy people and IBD patients. Based on these findings, a very logical theory has been proposed that IBD occurs due to changes in the composition of the gut microbiome that is considered normal [5]. In the intestinal tract of UC models and patients induced by the chemical DSS, bacterial diversity involved in maintaining normal long-term function was reduced, fungal populations increased, and methanogen diversity was reduced. This process leads to the overproduction of cytokines such as interleukin-6 (IL-6) and tumor necrosis factor-α (TNF-α), activates other inflammatory cells such as vascular endothelial cells, fibroblasts, etc., and promotes inflammation [6,7]. The dextran sodium sulfate (DSS)-induced colitis model is one of the most commonly used mouse models for IBD research [8]. DSS is a negatively charged sulfated polysaccharide that acts as a chemical colitogen with anticoagulant properties. When administered to mice, it damages the epithelial cells lining the colon, leading to inflammation and ulceration similar to IBD in humans. This model has many advantages, including ease of use, reproducibility, and the ability to control the extent and severity of inflammation by varying the concentration and duration of DSS treatment.

Recently, various factors, such as genetic/environmental factors, infectious factors, and disruption of the normal intestinal microbial bacterium, have been accumulated and combined as the etiology of IBD, resulting in an overreaction of the mucosal immune system that exceeds the threshold [9]. Current therapeutic agents (5-ASA, steroids, immunosuppressants, biologics, etc.) used for inflammatory bowel disease are mostly symptom-modulating and anti-inflammatory drugs, and in a situation where there is no complete cure, complications occur [10]. Due to the lack of efficacy and safety due to the disease, there is a need for continuous management and treatment that takes into account the patient’s health and constitution. Glucans are classified into alpha-glucans and beta-glucans. Alpha-glucans are generally found in plant starch, and beta-glucans are known to be present in the cell walls of cereals, mushrooms, and yeast [11]. Unlike alpha-glucans, beta-glucans exhibit non-specific immune reactions, and when absorbed into the digestive system, they are known to dramatically enhance the body’s immune function. Beta-glucans have attracted attention as a natural immunomodulator by enhancing immune function [12]. Among the natural extracts that enhance immunity, β-glucan exhibits anti-inflammatory, anti-cancer, and immunoregulatory effects [13]. The Polycan used in this study is a type of β-glucan derived from *Aureobasidium pullulans* SM-2001. It is a complex of bioactive ingredients including polysaccharides, proteins, organic acids, and vitamins. It is functional (KFDA health functional food individual certification type raw material certification) and its safety (FDA GRAS certification) is verified. β-1,3/1,6-glucan extracted from yeast cells, lactic acid bacteria cells, *Aureobasidium* sp. cells, etc. has excellent safety and is a substance with high utility value as a food material. In particular, the Polycan used in this study produces β-1,3/1,6-glucan extracellularly using an *A. pullulans* strain. This is a water-soluble β-1,3/1,6-glucan with high physiological activity. This is because, unlike common β-1,3-glucan, β-1,3/1,6-glucan produced by *A. pullulans* has unique physiological activity not found in plant polymers and has comparable physical properties [14]. It effectively promotes bone formation and bone loss [15,16,17]. Recently, studies have shown that Polycan can reduce the expression of inflammatory cytokines in osteoarthritis models and atopy models [18]. In this study, we investigated the anti-inflammatory and intestinal barrier effects of Polycan extract isolated from *A. pullulans* using a DSS-induced IBD model.

## 2. Results

### 2.1. Effects of Polycan of the Physiological Efficacy on DSS-Induced Colitis Mice

We examined the protective effect of Polycan against severe symptoms of colitis caused by DSS. Compared with the vehicle group, DSS-treated mice had significantly reduced body weight (Figure 1A). However, Polycan-treated mice had significantly improved weight loss compared to DSS-treated mice (Figure 1A,B). In addition to changes in body weight, the Polycan treatment significantly reduced the disease activity index compared to the DSS-alone treatment (Figure 1C). Moreover, colon length reduction by DSS was clearly increased by the Polycan treatment compared to the DSS-alone treatment (Figure 1D,E). Figure 1F shows the serum FITC-dextran content in the Polycan-treated group (139.33 ± 25 or 96.50 ± 11.07) compared to the normal group (25 ± 1.74) and the DSS-treated group (157.67 ± 25.79). 

### 2.2. Effects of Polycan on the Histopathological Changes of DSS-Induced Colitis Mice

Colonoscopies and H&E and PAS staining were performed to examine the efficacy of Polycan on colonic morphological and histological changes in a model of inflammatory colitis induced by DSS. The DSS-treated group (Veh group), showed inflammation-induced ulceration and inflammatory cell infiltration in the colonic tissue and crypto atrophy in colon tissues (Figure 2A,B; endoscopy and H&E staining). However, these histological and morphological changes were repaired by the Polycan treatment on day 14 (Figure 2A,B). Furthermore, the number of goblet cells in the DSS-treated group was significantly reduced (Figure 2C), which prevented the loss of remnant cells compared to the DSS-treated group during the Polycan treatment. These results indicate that Polycan can be used to ameliorate DSS-induced colitis (Figure 2).

### 2.3. Effects of Polycan of Proinflammatory Cytokines in DSS-Induced Colitis Mice

It has been reported that there is a close relationship between colitis and proinflammatory cytokines, and this was also investigated in this study. Thus, we measured the serum or RNA levels of TNF-α, IL-6, and IL-1β (pg/mL). The serum levels of TNF-α, IL-6, and IL-1β were significantly higher in the DSS-treated group than in the CTL group (Figure 3A–C). However, the serum levels of IL-6, TNF-α, and IL-1β in mice treated with Polycan were lower than those in the DSS-treated group (Veh). Concomitantly, the expression of RNA, as well as serum inflammatory cytokines, was clearly reduced by the Polycan treatment compared to the DSS treatment (Figure 3D–F).

### 2.4. Changes in the Expression of Genes Associated with Intestinal Barrier Function, Apoptosis Markers, and Tight Junction Proteins in DSS-Induced Colitis Mice

The effects of Polycan on the expression of apoptosis markers (e.g., caspase-3, PARP, Bcl-2 associated X [15], and B-cell lymphoma 2 [15]) and tight junction proteins (e.g., ZO-1, occludin, and claudin-2) were evaluated by using RNA and protein analyses. The expression of caspase-3, PARP, Bax, and Bcl-2 was increased in the DSS-treated group. In addition, when the Bax/Bcl-2 ratios after treatment with Polycan extract were compared with those in the CTL group, the expression of apoptosis-related RNA was recovered in a concentration-dependent manner (Figure 4A–C). To confirm the extent of inflammation in inflammatory bowel disease, we examined the level of mRNA for F4/80, a macrophage maker. Macrophage expression increased in the DSS-treated group compared with the control group but significantly decreased in the high-concentration treatment group with the Polycan treatment compared with the DSS-treated group (Figure 4D).

ZO-1 and occludin expression were significantly decreased, but claudin-2 expression was increased in the colon tissue of DSS-treated mice. However, the expression of these proteins was recovered by Polycan treatments (Figure 4E,F). Moreover, Western blot analysis showed that the decreased expression of ZO-1 and occludin in DSS-induced colitis was recovered by the Polycan treatment (Figure 4F). Taken together, these results allow Polycan to modulate tight junction protein levels upon Polycan treatment in a DSS-induced model of acute colitis model, which indicates that it may be relevant for therapeutic and preventive mechanisms of colitis.

### 2.5. Effects of Polycan on Apoptosis in DSS-Induced Colitis Mice

TUNEL assay and Western blot analysis were performed using colon tissue of DSS-induced colitis mice to determine the protective effects of Polycan on apoptosis. As expected, apoptosis induction was greatly increased by DSS treatment; however, Polycan effectively suppressed this effect (Figure 5A). DSS significantly increased the expression of apoptotic proteins compared to the CTL group. However, the Polycan-treated group markedly suppressed this apoptosis (Figure 5B–D).

### 2.6. Effects of Polycan on Decreasing MPO Activity in DSS-Induced Colitis Mice

To confirm the effect of Polycan on MPO-mediated oxidative damage, colon tissue was used to measure the chlorination and peroxidation activities. The results showed that DSS-treated mice had significantly higher MPO-mediated oxidative injury, chlorination, and peroxidation activity compared to the CTL group (Figure 6A,B). On the other hand, the activity increased by DSS was decreased by the Polycan treatment. These results indicated that Polycan could reduce oxidative stress by controlling chloride concentration and reducing substrate.

### 2.7. Effects of Polycan on Intestinal Microbial Composition

The microbial composition and proportions of each sample were determined at various taxonomic levels. Two major microbial phyla, Bacteroidetes and Firmicutes, accounted for the majority in this experiment. Depending on the relative proportions of the microbial community, there was significant clustering into two groups of control and DSS-treated conditions (Groups 1 and 2). Group 2 was clustered into two subgroups. That is, the DSS- and Polycan-treated groups were classified as Group 3. The 5-ASA group was classified into a different subgroup: Group 4 (Figure 7A).

Akkermansiaceae is a microorganism that produces acetic acid, which is the most abundant short-chain fatty acid in the intestine. When the gut microbiota at the genus level was altered, DSS administration at the door level clearly decreased the abundance of Bacteroidetes and Campylobacterota and increased the abundance of Firmicutes, Proteobacteria, and Verrucomicrobiota (Figure 7B). In addition, DSS administration greatly increased the abundance of Escherichia, a harmful bacterium, but the administration of Polycan reversed this effect. Analysis of the total microbial alpha diversity at the genus level showed that the alpha diversity increased with DSS treatment (Figure 7C). This finding indicates that DSS caused significant changes in normal intestinal bacteria.

## 3. Discussion

IBD is an intractable disease, and no drug has been developed to cure it to date [18,19]. Therefore, there is a need to prevent it or find new treatments. Patients with ulcerative colitis generally experience bloody stool or diarrhea mixed with mucus several times a day, suffer from fever or abdominal pain, show inflammation and ulcers in the large intestine, experience a reduction in the length of the rectum, and have inflammatory cells infiltrating into the injured area of the large intestine [18,19,20,21]. 

Since the cause of ulcerative colitis has not been identified and there is no fundamental treatment [22], the current treatment method is to relieve symptoms and maintain remission using aminosalicylic acid and steroids. Drug treatment for ulcerative colitis includes 5-aminosalicylic acid (5-ASA) preparations, corticosteroids, and immunomodulatory drugs. Statistics show that 20–40% of patients undergo colectomy due to ineffective drug treatment or side effects [23]. The surgical method is complicated and the aftereffects from the surgery are significant. Recent research trends are focused on minimizing side effects from natural substances, including natural material, and developing effective candidate drugs for ulcerative colitis [23,24].

In this study, Polycan demonstrated anti-inflammatory and protective effects on tight junction against DSS-induced colitis. In other words, Polycan has a protective effect on intestinal length shortening with weight loss in DSS-induced colitis mice and intestinal permeability, by regulating adhesion proteins (e.g., ZO-1, occludin, and claudin-2). This indicates that Polycan has an ameliorative effect on inflammatory bowel dysfunction. Infectious diseases related to the cause of colitis cytokine activate the protective response of immune cells such as macrophages, neutrophils, and lymphocytes in the colonic mucosa and induce TNF-α, IFN-γ, IL-6, and other immune cells [10]. DSS promotes inflammation by inducing excessive production of cytokines [25,26]. Our study showed that Polycan reduced serum levels of TNF-α, IL-6, and IL-1B. Furthermore, it reduced the activation of MPO, a factor associated with oxidative stress, in a DSS-induced colitis model, suppressing oxidative stress and macrophage infiltration. These data thus demonstrated that Polycan balances the secretion of inflammatory factors, regulates intestinal permeability associated with anti-inflammatory effects, and attenuates colitis induced by DSS.

A number of recent studies have shown that UC is associated with repetitive and persistent inflammatory responses associated with disruption of the intestinal epithelial barrier [23,27]. Apoptosis of intestinal epithelial cells, loss of function, and dysfunction of tight junction proteins can impair barrier function, which increases intestinal permeability [28,29]. As a result, colonic inflammation occurs with antigen and bacterial invasion, and tight junction degradation develops with the release of pro- and anti-inflammatory factors [30]. Treatment of mucosal epithelium with DSS damages the zonula occludens between mucosal epithelial cells and reduces the distribution of zonula occludens (ZO)-1, an intercellular junction. When the zonula occludens is damaged, intestinal permeability increases and induces an inflammatory response in the intestine, resulting in ulcerative colitis [31]. In addition, DSS causes excessive infiltration of granulocytes into mucous tissue, causing hypertrophy, ulceration around the epidermis, and severe edema in the tissue wall [32].

In the present study, we also confirmed the disruption of tight junction proteins along with epithelial cell apoptosis in DSS-treated mice. The administration of Polycan significantly inhibited colonic cell death and protected the expression of ZO-1, claudin-2, and occludin. Moreover, Polycan repaired the expression of tight junction proteins and their associated proteins and protected against apoptosis.

Several reports showed that the DSS-induced acute colitis model has increased numbers of lymphocytes, macrophages, and neutrophils [33]. Among them, macrophages play an essential role in the production of inflammatory cytokines and mediate the secretion of TNF-α, IL-6, and IL-1β [9,24]. In the present study, F4/80 staining showed that DSS-treated mice had increased macrophage infiltration, whereas Polycan-treated mice had decreased macrophage infiltration. The results showed that Polycan prevented macrophage infiltration into intestinal crypts and reduced inflammation by controlling inflammatory factors in the colon of DSS-induced colitis mice.

Although the pathogenesis of IBD is not yet clearly defined, various environmental factors are brought about by distorting mucosal immune responses [34]. A DSS-induced colitis model can be used to investigate the pathological mechanism of IBD [35,36,37]. DSS causes direct toxicity to epithelial cells, increases colonic mucosal permeability with complete loss of epithelial segments, allows direct contact between gut microbiota and host cells, and causes intestinal inflammation. An imbalance of gut microbiota, therefore, exacerbates pathological responses in the gut [8,38]. This study also showed that the ratio of Akkermansiaceae decreased in the DSS-treated group compared with the normal group, and increased in a dose-dependent manner when treated with Polycan. These results are consistent with the reduction and increase of acetate by DSS and Polycan treatment, respectively, suggesting that Polycan contributes to gut health through the production of short-chain fatty acids.

## 4. Materials and Methods

### 4.1. Chemicals and Reagents

DSS, 5-aminosalicylic acid (5-ASA), and hematoxylin and eosin (H&E) solutions were purchased from MP Biomedicals (Santa Ana, CA, USA) and Sigma Aldrich (St Louis, MO, USA). The enzyme-linked immunosorbent assay (ELISA) kits for determining myeloperoxidase (MPO) activity, TNF-α, IL-1β, and IL-6 were purchased from Thermo Fisher Scientific (Waltham, MA, USA) and eBioscience (San Diego, CA, USA). RIPA lysis buffer and phosphatase and protease inhibitor cocktails were obtained from Millipore (Darmstadt, Germany) and Roche (Basel, Switzerland). The BCA protein quantification kit, fluorescence-tagged antibody, anti-Zonula occludens-1 (ZO-1) antibody, anti-occludin antibody, and anti-F4/80 antibody were purchased from Thermo Fisher Scientific and Santa Cruz Biotechnology (Dallas, TX, USA).

### 4.2. Preparation of Polycan

The Polycan used in this study was β-glucan fermented with *A. pullulans* SM-2001, which was provided by Glucan Co., Ltd (Jinju-si, Korea). Hereinafter, the extract is referred to as “Polycan”. β-1,3/1,6-glucan (β-1,3 linkage 68: β-1,6 linkage 32) has an average molecular weight of 2.6 × 105 Da. The Polycan was used to evaluate its efficacy on DSS-induced colitis. Polycan prepared immediately before the experiment was filtered through a syringe filter (PTFE, 0.22 µM, 25 mm, Sterlitech Co., Ltd., Auburn, WA, USA) and administered to the animals.

### 4.3. Mice and Treatment

All animals used in the experiments were seven-week-old male C57/BL6 mice purchased from DooYeol Biotech (Seoul, Republic of Korea). The experiment proceeded after the mice underwent a 7-day adaptation period in the animal laboratory of the Korea Institute of Oriental Medicine (KIOM). Water and food intake were not restricted during the adaptation period. The animals were provided with a standardized environment and a 12-h day-and-night schedule was maintained. Room temperature (23 °C ± 2 °C) and humidity (50–55%) were maintained at titratable levels. All experiments were performed after approval by the Korea Institute of Oriental Medicine Institutional Animal Care and Use Committee (KIOM-22-064). They were divided into five groups (n = 10 per group): vehicle-treated control (CTL), 5% DSS (Veh), 5% DSS + Polycan 250 mg/kg (Polycan 250), 5% DSS + Polycan 500 mg/kg (Polycan 500), and 5% DSS + 5-ASA 100 mg/kg (5-ASA). During the experiments, the body weight of the mice was measured daily before the oral administration of Polycan (Figure 8).

### 4.4. Analysis of Epithelial Paracellular Permeability

After 14 days of experimentation, the mice were examined for epithelial pericellular permeability using a non-absorbable FITC (fluorescein isothiocyanate)-conjugated dextran probe (FD-4). Briefly, the mice fasted for 8 h at the end of the experiment and were orally dosed with 400 μg of fluorescein isothiocyanate (FITC)-dextran (4 kDa, Sigma-Aldrich, St. Louis, MO, USA)/g body weight. After 4 h, after the separation of serum from blood, intestinal permeability was measured at wavelengths of 490 nm and 520 nm emission wavelengths (VERSAmax microplate reader, Molecular Devices).

### 4.5. Endoscopy of the Large Intestine and Histological Analysis

After the end of the experiment, the mice were anesthetized with isoflurane and endoscopically examined using a mini-endoscope (OLYMPUS, Tokyo, Japan, length 670 mm, diameter 2.8 mm) to observe histological changes in the colons. The animals were then sacrificed for intestinal tissue after collecting whole blood from the abdominal vein. Isolated colons were fixed with a 4% paraformaldehyde solution, embedded in paraffin blocks, and sectioned using a microtome. To observe histological changes in the colon, an H&E and a PAS (periodic acid-Schiff) solution were used.

### 4.6. Determination of Serum Cytokines of ELISA and MPO Activity

Whole blood from the abdominal vein of the mice was incubated overnight at 4 °C and then centrifuged (3000× *g* at 4 °C) for 15 min to isolate the serum. The isolated serum was stored at −80 °C until use. Serum IL-1β, IL-6, and TNF-α levels were determined using ELISA kits according to the manufacturer’s instructions. In addition, MPO, a neutrophil infiltration marker that is activated when inflammation occurs as an enzyme in neutrophils, was measured using an MPO activity assay kit.

### 4.7. Terminal Deoxynucleotidyl Transferase dUTP Nick End Labeling (TUNEL) Assay

To analyze the extent of cell death induced by Cortisol, apoptotic cells were stained with the TUNEL method. That is, paraffin tissue sections were stained using the TUNEL apoptosis detection kit (Beyotime Biotechnology, Beijing, China), and nuclei were stained with 4′,6-diamidino-2-phenylindole (Sigma, St. Louis, MO, USA) dye.

### 4.8. Real-Time Polymerase Chain Reaction (PCR) of Colon Tissue

The colon tissues of 0.1 g were immediately frozen in liquid nitrogen and homogenized after adding 1 mL of TRIzol (Life Technologies, Carlsbad, CA, USA). Isolated total RNA was stored in a −80 °C deep freezer until analysis and cDNA was synthesized with 1 μg total RNA using the RevertAid RT reverse transcription kit (Thermo Fisher Scientific, Waltham, MA, USA). The synthesized cDNA was diluted with DEPC-treated water, stored at −20 °C, and used as a template for real-time PCR. mRNA expression analysis was performed (Appendix A).

### 4.9. Western Blot Analysis

Colon tissue total protein was prepared in a RIPA buffer (0.1% SDS, 1% Triton X-100, 0.5% sodium deoxycholate, 50 mM Tris, 150 mM NaCl, 50 mM NaF, 0.5 M EDTA, 0.1 M EG), and mouse intestinal proteins were extracted using a BCA kit. After protein quantification, the same proteins were separated by 10% and 12% SDS-PAGE and transferred to membranes, followed by 0.05% Tris-buffered saline. After blocking with 5% BSA in Tween 20, the primary antibody is O/N incubated at 4 °C with antibody (Cell Signaling, Danvers, MA, USA) diluted 1:1000. The next day, after washing with 0.1% TBST 3 times for 10 min, anti-rabbit secondary antibody diluted at a ratio of 1:1000 was used as the secondary antibody and incubated at room temperature for 1 h. Detection was performed using the Imaging System (Bio-Rad, Hercules, CA, USA).

### 4.10. Feces Collection and 16S rRNA Sequencing of Intestinal Microbiota Analysis

Fecal collection and gut microbiota analysis were performed according to the methods [39]. Fresh feces were placed into a sterilized centrifuge tube (1.5 mL) and stored at −80 °C. A library for 16S metagenomic sequencing was prepared by amplifying the specific primer (515F-806R) of 16S rRNA using the Hercules kit on the Illumina platform to construct a library of DNA extracted from fecal samples. Operational taxonomic units were calculated at a 97% sequence identity cutoff using the VSEARCH algorithm, and the taxonomic assignment was performed using the RDP classifier (https://rdp.cme.msu.edu/classifier/ (accessed on 20 February 2023). α-Diversity and heatmap clustering analysis was performed using MicrobiomeAnalyst (https://www.microbiomeanalyst.ca/Microbiome-Analyst/home.xhtml, accessed on 20 February 2023).

### 4.11. Statistical Analysis

GraphPad Prism version 5 was used to conduct statistical analyses and draw graphs. Experimental values are presented as means ± standard error of the mean. One-way analysis of variance (ANOVA) was used to determine the significant difference. Statistical significance was considered at *p* < 0.05. Post hoc comparison of means was conducted using Tukey’s test (for one-way ANOVA) or Bonferroni’s test (for two-way ANOVA: DSS treatment, Polycan treatment, and interaction between DSS and Polycan administration) for multiple comparisons, when appropriate.

## 5. Conclusions

In this study, we confirmed the alleviating effects of Polycan through regulation of the intestinal barrier function and anti-inflammation in an ulcerative colitis model. To confirm the anti-colitis efficacy of Polycan in a DSS-induced colitis mouse model, clinical symptoms and inflammatory indicators of colitis were analyzed after orally administering Polycan along with 5% DSS to mice for 14 days. As a result, the administration of Polycan was found to improve indicators of murine colitis symptoms induced by DSS, namely, disease activity including weight loss and bloody diarrhea, colonic mucosal proliferation, and increased inflammatory cytokines levels. Additionally, damage to the colonic mucosa and intestinal barrier worsened by DSS was also reduced in the Polycan-administration group. These results suggest that Polycan has anti-colitis efficacy. However, if the efficacy and related mechanisms of the bioactive substances contained in Polycan are identified, it is expected that can be developed as a health food for preventing or improving colitis.

## Figures and Tables

**Figure 1 ijms-24-14773-f001:**
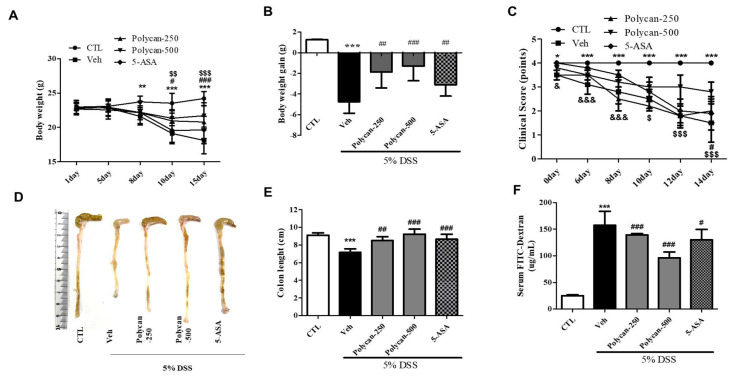
Effects of Polycan on body weight, colon length, clinical score, and serum fluorescein isothiocyanate (FITC)-dextran permeability in DSS-induced colitis model. (**A**,**B**) Body weight, (**C**) clinical score, (**D**,**E**) colon length, and (**F**) serum FITC-dextran permeability. Mice were orally administered with Polycan or 5-aminosalicylic acid (5-ASA) (100 mg/kg) before DSS treatment. Body weights were monitored before Polycan or 5-ASA treatment. Colonic lengths were measured after excision from euthanized mice. Epithelial paracellular permeability was measured using an FITC-conjugated dextran probe. Results represent the mean ± standard error of the mean of each mouse in the same group. The figure (**A**) and (**C**): * *p* < 0.05, ** *p* < 0.01 and *** *p* < 0.001 CTL vs. Veh; ^#^
*p* < 0.05 and ^###^
*p* < 0.001 Veh vs. Polycan 250; ^$$^ *p* < 0.01, and ^$$$^ *p* < 0.001 veh vs. Polycan 500. ^&^ *p* < 0.01 and ^&&&^ *p* < 0.001 Veh vs. 5-ASA. All other figure (**B**,**E**,**F**): *** *p* < 0.001 versus the control group (CTL), ^#^ *p* < 0.05, ^##^ *p* < 0.01, and ^###^ *p* < 0.001 versus the DSS-treated group (Veh).

**Figure 2 ijms-24-14773-f002:**
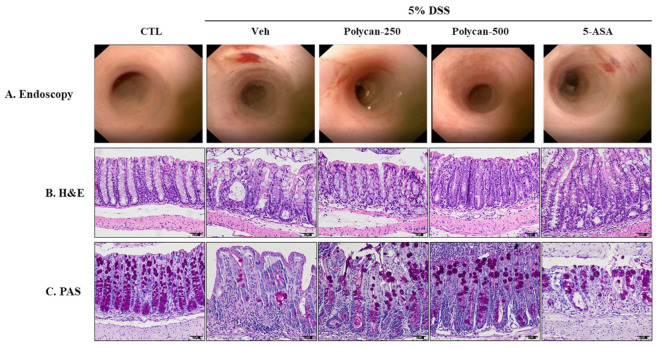
Protective effect of Polycan on histopathological changes in DSS-induced colitis model. (**A**) Representative image of endoscopic colon tissue, (**B**) hematoxylin and eosin (H&E) staining (magnification 400×), and (**C**) PAS staining (magnification 400×). On day 14, mucosal damage was confirmed via mini-endoscopy.

**Figure 3 ijms-24-14773-f003:**
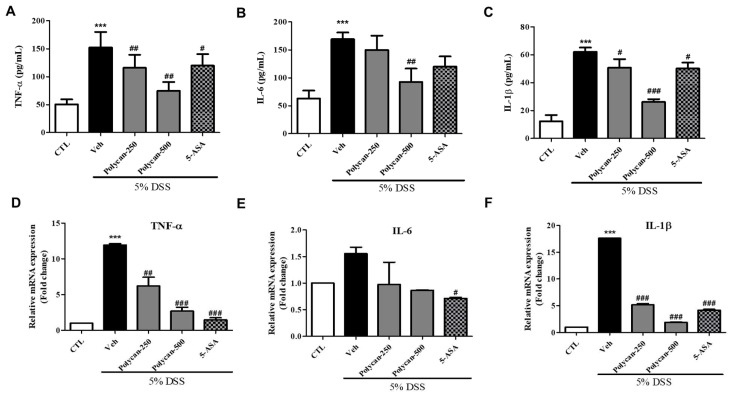
Effects of Polycan on the serum and mRNA levels of proinflammatory cytokines in DSS-induced colitis model. Serum levels of (**A**) tumor necrosis factor (TNF)-α, (**B**) interleukin (IL)-6, and (**C**) IL-1β. The serum levels of proinflammatory cytokines were detected using an enzyme-linked immunosorbent assay kit. Relative expression of mRNA for (**D**) TNF-α, (**E**) IL-6, and (**F**) IL-1β. Cytokine and mRNA levels are presented as the mean ± standard error of the mean of three independent experiments. *** *p* < 0.001 versus the control group, ^#^
*p* < 0.05, ^##^
*p* < 0.01, and ^###^
*p* < 0.001 versus the DSS-treated group.

**Figure 4 ijms-24-14773-f004:**
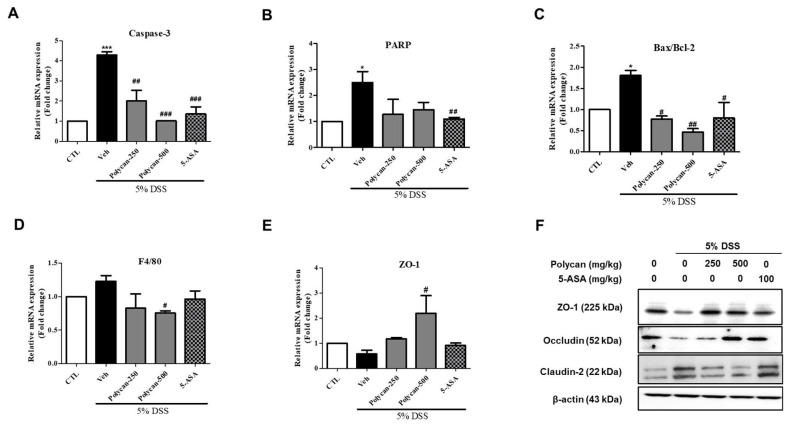
Protective effect of Polycan on apoptosis and tight junction in DSS-induced colitis model. (**A**,**B**) Representative mRNA expression of caspase-3 and PARP. (**C**) Ratio of Bax/Bcl-2 mRNA expression. (**D**) Representative mRNA expression of F4/80, a macrophage maker. (**E**) Representative mRNA expression of ZO-1, a tight junction factor. (**F**) Western blot analysis of the intestinal tissues of ZO-1, occludin, and claudin-2. Values are presented as the mean ± standard error of the mean of three independent experiments. * *p* < 0.05, *** *p* < 0.001 versus the control group, ^#^
*p* < 0.05, ^##^
*p* < 0.01, and ^###^
*p* < 0.001 versus the DSS-treated group.

**Figure 5 ijms-24-14773-f005:**
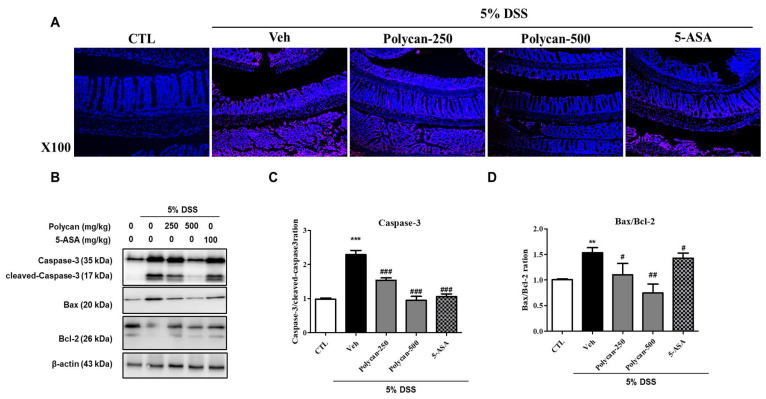
Protective effect of Polycan on apoptosis induction in the DSS-induced colitis model. (**A**) Representative image of TUNEL staining. (**B**) Representative expression of caspase-3, Bax, and Bcl-2 proteins determined by Western blot analysis. β-actin was used as the protein loading control. (**C**) The ratios of caspase-3/cleaved-caspase-3 and (**D**) Bax/Bcl-2 were determined by Western blot analysis. Values are expressed as the mean ± standard error of the mean of three independent experiments. ** *p* < 0.01, *** *p* < 0.001 versus the control group, ^#^
*p* < 0.05, ^##^
*p* < 0.01, and ^###^
*p* < 0.001 versus the DSS-treated group.

**Figure 6 ijms-24-14773-f006:**
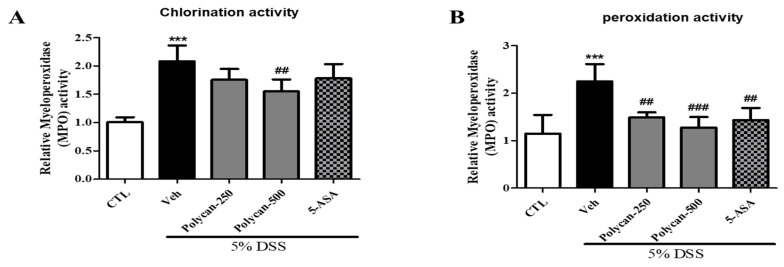
Effects of Polycan on myeloperoxidase (MPO) activity in the DSS-induced colitis model. (**A**) Chlorination activity. (**B**) Peroxidation activity. Mice were orally administered with Polycan (250, 500 mg/kg) or 5-aminosalicylic acid (5-ASA) (100 mg/kg) before DSS treatment. The MPO activity was measured using an MPO activity assay kit according to the manufacturer’s instructions. Data are presented as the mean ± standard deviation (n = 10) from triplicate experiments. Different superscripts are significantly different at *p* < 0.05 by Duncan’s multiple range test. *** *p* < 0.001 versus the CTL group. ^##^
*p* < 0.001, ^###^
*p* < 0.01 versus the DSS group.

**Figure 7 ijms-24-14773-f007:**
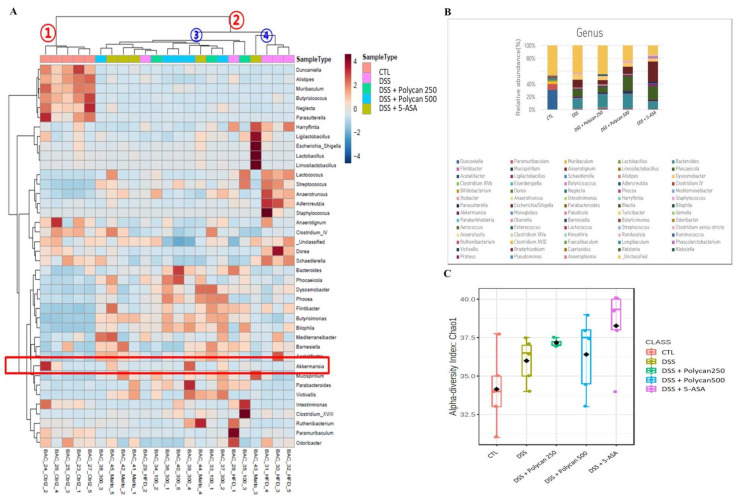
Effect of Polycan on relative gut bacterial levels in dextran sodium sulfate (DSS)-induced colitis model. (**A**) Heatmap clustering of genus level intestinal flora in DSS-induced colitis model. (**B**) The microbiota composition and proportion among five groups at the genus in the DSS-induced colitis model. (**C**) Genus level α-diversity analysis in DSS-induced colitis model.

**Figure 8 ijms-24-14773-f008:**
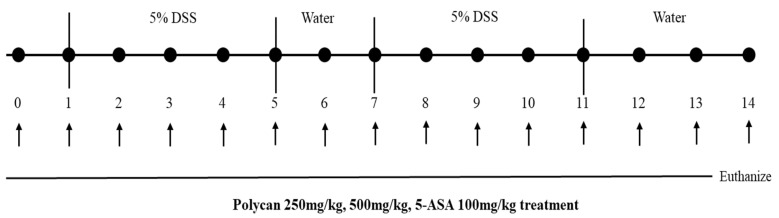
Animal model of DSS-induced colitis.

## Data Availability

The data presented in this study are available upon request from the corresponding author.

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
