# Peer review of "Effect of Polycan, a β-Glucan from Aureobasidium pullulans SM-2001, on Inflammatory Response and Intestinal Barrier Function in DSS-Induced Ulcerative Colitis"

_ijms, 2023, doi:10.3390/ijms241914773_

Round 1

Reviewer 1 Report

The manuscript "Effect of Polycan, a β-glucan from Aureobasidium
pullulans SM-2001, on inflammatory response and intestinal barrier function
in DSS-induced ulcerative colitis" is well described, structured and has demonstrated
the effect of polycan against colitis (DSS-induced model) in several aspects such
as physiological conditions, inflammatory cytokines, barrier function, apoptosis,
markers, and tight junction proteins and intestinal microbial composition.
I recommend it for publication after minor changes.

1. Please consider to include information about the intestinal microbial composition (UC)
in the abstract and introduction.
2. Line 88: Figure citation is wrong, if the topic is mice body weight, it is described
in fig 1A instead 2A.
123. The first sentence in this line makes no sense.

Author Response

  1. Please consider to include information about the intestinal microbial composition (UC) in the abstract and introduction.

Response: Thank you so much for your time and consideration, as well as your positive feedback. As you pointed out, I have modified the Abstract and Introduction section as you pointed out. (Abstract, Line 30 and Introduction, Line 51)

  1. Line 88: Figure citation is wrong, if the topic is mice body weight, it is described in fig 1A instead 2A. 123. The first sentence in this line makes no sense.

Response: Thank you so much for your time and consideration, as well as your positive feedback. As you pointed out, I have modified Result section. Result (Line 101).

Reviewer 2 Report

The manuscript by Hyun Ju Do studied the effect of polycan, a β-glucan from Aureobasidium pullulans SM-2001, on dextran sodium sulfate (DSS)-induced colitis in mice. The authors showed that oral administration of polycan protected intestinal barrier function and inhibited inflammation and apoptosis in DSS-induced colitis. Overall, the manuscript is well written and organized, and presents the important data that support the polycan as a suitable candidate for improving gut health. Methodology used are also appropriate. Thus, I would like to recommend the manuscript for publication after minor revision.

Specific comments:

Many studies demonstrated the beneficial effects of  b-glucan dietary supplementation (e.g., 1,3- β-glucan, oat β-glucan) in DSS-induced colitis rodent models. The authors should discuss what are the advantages of polycan over other  β-glucan on gut health.

The authors did not describe the Figure 4D in the main text. Please explain.

The authors showed that DSS administration decreased the abundance of Bacteroidetes and Campylobacterota and increased the abundance of Firmicutes, Proteobacteria, and Verrucomicrobiota (Figure 7B). Did polycan treatment reverse these changes?

The authors showed that total microbial alpha diversity was increased with DSS administration (Figure 7C). Did polycan treatment produce significant effect on the alpha diversity of gut microbiota?

Nil

Author Response

Many studies demonstrated the beneficial effects of b-glucan dietary supplementation (e.g., 1,3- β-glucan, oat β-glucan) in DSS-induced colitis rodent models. The authors should discuss what are the advantages of polycan over other β-glucan on gut health.

Response: Thank you so much for your time and consideration, as well as your positive feedback. As you pointed out, I have modified Introduction section. (Introduction, Line 86).

The authors did not describe the Figure 4D in the main text. Please explain.

Response: Thank you so much for your time and consideration, as well as your positive feedback. As you pointed out, I have modified Result section. (Result, Line 160).

The authors showed that DSS administration decreased the abundance of Bacteroidetes and Campylobacterota and increased the abundance of Firmicutes, Proteobacteria, and Verrucomicrobiota (Figure 7B). Did polycan treatment reverse these changes?

The authors showed that total microbial alpha diversity was increased with DSS administration (Figure 7C). Did polycan treatment produce significant effect on the alpha diversity of gut microbiota?

Response: Thank you so much for your time and consideration, as well as your positive feedback. As you pointed out, answer to your question is here.

Overall, in the DSS model, acetic acid is the most abundant short-chain fatty acid in the intestines, and most anaerobic microorganisms in the large intestine produce acetic acid, with species such as Bacteroides and Akkermansia producing acetic acid and propionic acid. In this study, the ratio of Akkermansiaceae decreased with DSS administration compared to the normal group, but tended to increase in a concentration-dependent manner when Polycan was administered. This trend is consistent with the tendency for acetic acid to decrease with DSS administration and increase with Polycan administration, and it can be assumed that polycan contributes to intestinal health through the production of acetic acid short-chain fatty acids through the total proliferation of Akkermansiaceae bacteria. Representative intestinal microorganisms that belong to harmful bacteria include Clostridium and Escherichia, and also include Klebsiella and Proteus. Proteus increased upon administration of DSS, but tended to decrease upon administration of Polycan. Escherichia, which belongs to harmful bacteria, increased significantly when administered, but showed a clear to decrease upon administration of Polycan.